# A Tactical Conflict Resolution Proposal for U-Space Zu Airspace Volumes

**DOI:** 10.3390/s21165649

**Published:** 2021-08-22

**Authors:** Jesús Jover, Aurelio Bermúdez, Rafael Casado

**Affiliations:** Albacete Research Institute of Informatics, University of Castilla–La Mancha (UCLM), Campus Universitario s/n, 02071 Albacete, Spain; jesus.jover@uclm.es (J.J.); rafael.casado@uclm.es (R.C.)

**Keywords:** urban air mobility (UAM), unmanned aerial vehicle (UAV), conflict detection and resolution

## Abstract

Conflict management between UAVs is one of the key aspects in developing future urban aerial mobility (UAM) spaces, such as the one proposed in U-Space. In the framework of tactical conflict management, i.e., with the UAVs in flight, this paper presents PCAN (Prediction-based Conflict-free Adaptive Navigation). This relatively simple navigation technique predicts the occurrence of the conflict and avoids it by modifying the velocity vector of the UAVs involved. The performance evaluation carried out demonstrates its effectiveness compared to similar techniques, even in high-density scenarios, while proving a low overhead in flight time or in the distance traveled by the UAVs to reach their destinations.

## 1. Introduction

In recent years, interest in Unmanned Aerial Vehicles (UAVs) [1] has grown considerably, given the wide range of possibilities they offer. Within this revolution, it is more than likely that, in the short or medium term, the airspace of the metropolitan environment will be shared by “traditional” manned aerial vehicles and UAVs, which will be mainly electric and will have vertical takeoff and landing capabilities (eVTOL) [2]. These UAVs will fly autonomously [3] at low or very low altitude levels, covering a wide range of services, including the transport of goods or people, and contributing to reducing the surface and sub-surface congestion and the carbon footprint produced by our daily activity. These scenarios are called Urban Air Mobility/Advanced Air Mobility (UAM/AAM) [4].

Several public and private organizations have begun to restructure the airspace to integrate UAVs into it. In this sense, the concept of UTM (Unmanned Aircraft System Traffic Management) has been developed in the United States [5,6], while in Europe, this initiative has adopted the name U-Space [7,8]. Similar initiatives have appeared in other geographical areas, such as Korea [9] and Australia [10].

Conflict management is one of the many technical challenges to be solved in this scenario. Here, a “conflict” is understood as a situation in which two or more UAVs are at a distance less than a minimum separation predetermined by regulation. This is a classic problem in traditional Air Traffic Management (ATM) [11], where it is called Conflict Detection and Resolution (CD&R), and there are already countless resolution proposals [12,13]. In short, the aim is to avoid the occurrence of a collision between two aircraft at all costs.

One possible way of classifying conflict management techniques is to differentiate between those that obtain a priori a set of conflict-free routes or flight plans for the UAVs and those that detect conflicts during flight and resolve them by modifying the flight plan of the UAVs involved. Along the same lines, both UTM and U-space distinguish between strategic (pre-flight) and tactical (in-flight) conflict management.

Within the second group of techniques, there are strategies based on the calculation of the space of valid velocities that prevent any potential conflict between UAVs, such as the popular ORCA (Optimal Reciprocal Collision Avoidance) algorithm [14] and, more recently, the BBCA (Bounding Box Collision Avoidance) algorithm [15]. Still, there are also numerous force-based strategies, such as APF (Artificial Potential Field) [16], or swarm intelligence algorithms, such as PSO (Particle Swarm Optimization) [17]. It is also worth mentioning the existence of a group of methods, referred to as “sense-and-avoid,” in which the UAV is equipped with special hardware (LiDAR type) that allows a fast response in case of imminent collision [18,19].

One of the main characteristics of U-Space is that all operations must be safe. In this sense, and from the perspective of tactical conflict management, this paper presents a new proposal for navigating autonomous UAVs in UAM scenarios that manages to avoid any conflict between them while executing their corresponding flight plans.

Our proposal, called PCAN (Prediction-based Conflict-free Adaptive Navigation), is based on predicting the conflict based on estimating the future position of the UAVs. To prevent the occurrence of each predicted conflict, PCAN proceeds to adapt the velocity vector of the UAVs involved, considering the airspace situation. PCAN works in a centralized way, from the position, velocity, and destination information of all UAVs flying over the airspace.

Apart from preventing the occurrence of any conflict between the UAVs in flight, the proposed algorithm has a low computational cost, which makes it very suitable for urban mobility environments with a high density of UAVs, where fast response to conflicts is demanded. Additionally, as we will proof, the overhead (in terms of flight time and distance traveled by the UAVs) of avoiding all conflict is more than reasonable.

The rest of the paper is structured as follows. Next, Section 2 describes the U-Space context, as well as different techniques for in-flight conflict management. Then, Section 3 focuses on the detailed description of our proposal. Section 4 includes an analysis of the PCAN algorithm in various scenarios and its comparison with similar proposals. Finally, Section 5 presents our conclusions and outlines future works in this line.

## 2. Background

### 2.1. Tactical Conflict Management in U-Space

As stated, U-Space is focused on the safe integration of many UAVs into the airspace. To this end, it plans to introduce a set of services that will be deployed progressively, based on 4 phases (U1, U2, U3, and U4) with an incremental complexity and level of integration [8,20].

U-Space services related to tactical conflict management are in phase U2. Very briefly, a tracking service provides the estimated positions and trajectories of the UAVs. Then, the monitoring service uses the above information and, if applicable, alerts on potential conflicts. The tactical conflict resolution service is responsible for modifying the UAV flight plan. Finally, the emergency management service provides the UAVs involved with the action to be taken to mitigate the risk.

For UAV/UAS flight, U-Space defines a new Very Low Level (VLL) airspace, which coexists with the ICAO (International Civil Airspace Organization) A-G airspaces [21]. The VLL airspace is decomposed into three volumes, X, Y, and Z, which offer different services and have different access requirements. The main difference between these volumes lies precisely in the provision of conflict resolution services.

There are no such services in X-type volumes, and the pilot, in whose visual range the UAV is located (this is known as Visual Line of Sight, or VLOS), is responsible for ensuring a safe operation. In Y-type volumes, Beyond Visual Line of Sight (BVLOS) flights are allowed using remote pilots connected to U-Space, with strategic (but not tactical) conflict resolution services. Operation plans must be approved prior to takeoff. The drawback is that this can result in large separations between aircraft, both in space and time.

The Z-type volume allows for a higher density of operations, adding tactical conflict resolution services, which reduce the residual risk of strategic for in- and out-of-visual-range and autonomous flights. Specifically, and as discussed above, these services should check for any potential conflict in real-time, based on the current position of all aircraft (and, if possible, their flight plans), and provide the necessary avoidance instructions to the aircraft involved in the form of changes in velocity, altitude or course. If the airspace is controlled by the Air Traffic Service (ATS), it is called a Za volume. As we assume in this paper, the airspace is controlled by U-Space. It is referred to as a Zu volume.

### 2.2. Conflict Resolution Techniques for UAV

As discussed, UAV conflict management techniques have traditionally been classified into proactive (or strategic) and reactive (or tactical). Other possible classifications consider whether conflict control is centralized or distributed in each UAV or whether there is communication between UAVs sharing the airspace [22].

Within strategic management, the literature is abundant in collision-free path planning mechanisms, which have been widely explored for years in mobile robotics. Some of the most popular ones are visibility graphs, Probabilistic Road Maps (PRM), Rapidly-exploring Random Trees (RRT), A-Star (A*) and its multiple variants, Ant Colony Optimization (ACO), Tabu Search (TS), and Voronoi diagrams [23]. In general, all these techniques share a high computational complexity. In the following, we describe with more detail some of the techniques for tactical UAV conflict management.

ORCA [14] is a velocity-based conflict avoidance method. These methods are based on choosing the optimal velocity for a mobile agent (in our case, a UAV) from among all those velocities that avoid conflict with other mobile agents (called “valid velocities”). Each agent predicts its future position and nearby agents based on current velocities and, under this assumption, determines its new velocity according to some optimization criterion. In ORCA, each agent computes a Velocity Obstacle (VO) with respect to each agent flying in its environment. The VO includes all the prohibited velocities for the UAV. ORCA guarantees collision-free navigation in sparse scenarios. In dense scenarios it is possible not to find a valid velocity. Linear programming techniques are used to solve this problem, with a medium to high computational cost [24].

BBCA [15] is also a velocity-based mechanism with lower complexity than ORCA and, therefore, is more appropriate for real-time conflict management. Basically, in BBCA the set of valid velocities for each UAV is represented by a rectangular area. This area is modified in each execution of the algorithm (by very simple operations) to avoid conflict with each of the nearby UAVs. BBCA can avoid any conflict in scenarios with two UAVs, and most of them in denser scenarios, with a reasonably acceptable overhead on the distance traveled and the final flight time of the UAVs.

On the other hand, APF [16] is an example of a force-based conflict avoidance method. These methods simulate particle systems, where each particle exerts a certain force on the nearest ones. In the case of APF, the motion of each agent is determined by an attractive force towards its final destination, while the rest of the agents behave as obstacles exerting repulsive forces on it [25]. The main drawback of these methods is the existence of local minima that prevents the agents from moving towards their goal and a high computational cost.

There are also methods based on swarm intelligence, such as PSO [17]. They are inspired by the behavior of certain animal species, organizing agents into groups that work together to obtain an optimal solution to the problem of reaching their destinations (while avoiding conflicts between them). In PSO, obtaining this optimal solution is based on a continuous optimization problem. The distance of each agent to its destination is iteratively calculated and shared with the rest of the agents. These algorithms also involve a high computational cost, not being good candidates for real-time applications with multiple agents.

Another recent proposal is [26], where conflict management is performed in three separate stages. The first stage consists of strategic path planning using a PSO-type algorithm. Later, in a “pre-tactical” stage, it is proposed to delay the start of flights to avoid unresolved conflicts in the previous stage. Finally, residual conflicts that may appear in flight are solved using the “hovering” technique, which stops the UAV in the air for the time necessary to avoid the conflict.

## 3. Conflict-Free Navigation

This section presents our proposal for conflict-free navigation in UAM scenarios, which we have named PCAN (Prediction-based Conflict-free Adaptive Navigation). PCAN aims to avoid any conflict between a set of UAVs in flight, regardless of the airspace configuration. As the strategy will be based on modifying the velocity vector of the UAVs, the algorithm should try to introduce as little change as possible in the optimal velocity of each UAV. We will understand as optimal a solution in which the UAVs fly in a straight line towards their destination, traveling at the maximum possible velocity (which we call $v_{max}$). We call this strategy the “direct” method or algorithm.

We will base the choice of the new velocity vector for each UAV on the future state of the airspace. Considering the future position of a set of UAVs, we will decide which ones should modify their velocity vector and which ones should not to reach a conflict-free solution. PCAN employs two strategies to modify the velocities: 1) modify the direction of the velocity vector with a certain angle and 2) modify the modulus of the velocity vector, reducing the flight velocity below the maximum, but maintaining the direction and sense. Both modifications will be made on the direct velocity, trying to alter it as little as possible.

From now on, ”airspace” refers to the region of the space where UAVs fly. We assume that all UAVs fly at the same altitude. Consequently, we tackle conflict detection in the 2D plane, working on an area where UAVs move from their initial position to their final (or destination) position. Nevertheless, it is possible to run the algorithm in parallel by using multiple layers at different heights. The criteria for the layer choice are left to the U-Space service provider and can be very varied (UAV heading, priority, maneuverability, airspace congestion...). The only requirement is that a UAV must participate in the conflict management of the layers below the deployment one during vertical takeoff and landing.

Before going into detail, we will introduce some starting definitions and assumptions.

### 3.1. UAV Dynamic Model

A comprehensive model of the dynamic behavior of a quadcopter-type UAV can be found in previous work [27]. In the present paper, it is sufficient to apply the simplified 2D model described below.

Let x=[p p˙] be the dynamic state of a UAV, where vectors p=[px py]∈ℙ and p˙=[vx vy]∈V, represent its current position and velocity, respectively. Let v=[vx vy]∈V be the commanded (or desired) velocity for that UAV. Then, its dynamic behavior is modeled by the following first order system: p¨=(v−p˙)/τ, being τ the response time of the system (time employed to achieve the 63% of the desired value). Note that, the first-order system applies to the velocity of the UAV. However, due to it has been described with respect to its position, a second-order derivative appears. Figure 1 shows an example.

Under these conditions, the position error experienced by the UAV due to the delay in following the commanded velocity is $e=\tau \left|v-{\dot{p}}_{0}\right|$. The maximum error would occur in a scenario with both vectors at maximum speed and opposite direction: $e_{max}=2\tau v_{max}$. As we will see later, the conflict detection mechanism will consider the error due to the dynamic behavior of the UAVs, consequently increasing their safety radius.

Let A={a1,t,a2,t…an,t} be the set of aircraft in the airspace at time *t*. Each element a=[x R]∈A represents the status vector of a UAV, including its dynamic behavior and a route composed of several waypoints to its destination. Let R={w1,w2…wm} be an ordered sequence of waypoints wi∈ℙ.

Without loss of generality, we assume that all UAVs behave homogeneously, initially flying at a predetermined maximum velocity $v_{max}$ and they are considering a safety radius r (which delimits their “protected zone”). Figure 2 shows an example.

With this definition, the optimal behavior for each UAV in ideal airspace (without restrictions due, for example, to the presence of buildings or no-fly zones) would be to fly in a straight line towards the next waypoint in its route, at a velocity $v_{max}$ (what we have called the direct method). Table 1 shows the implementation of this navigation system, which is executed periodically. If the UAV would reach the current waypoint before the next execution (02), it switches to the next waypoint in route (03–04). If multiple UAVs behave in this way in the same airspace, they may collide in flight.

From now on, the expression “navigation computation” refers to the operation consisting in assigning to each UAV in the airspace a velocity with which it must move in order not to cause any conflict. A “valid velocity” for a UAV is a velocity that does not produce any conflict between it and the rest of the UAVs in the airspace. “Final velocity” is the valid velocity that each UAV will use to move after the execution of the navigation computation.

UAV movement through airspace occurs in given units of time (e.g., seconds). We assume that all displacements start simultaneously, but not all of them end at the same time. This will depend on the initial and final positions of the UAVs, their velocities during the travel, and the route they take. Navigation computation also occurs in units of time, with the same unit as UAV movement. We assume that navigation is executed every t units of travel. We call this parameter $t_{nav}$. In other words, $t_{nav}$ defines how many time units a UAV can travel with the same navigation computation.

### 3.2. Conflict Prediction Mechanism

In this subsection, we will first detail a mechanism to check for the occurrence of a future conflict between two UAVs. From this, we will describe how to predict whether a given UAV will be involved in a conflict with any other UAV present in the airspace.

If we assume that a UAV $a_{i}$ maintains its velocity constant, then its position at a future time t (relative to the current time) is provided by Equation (1):(1)$$p_{i,t}=p_{i,0}+v_{i}t$$

Two UAVs, $a_{1}$ and $a_{2}$, present a conflict if $\left|\bar{p_{1}p_{2}}\right|<2r$. The system of equations shown in Equation (2) allows us to determine the existence of a conflict between them, obtaining the instants and positions of its beginning and end.
(2)$$\{\begin{array}{ll} \;\;p_{1,t} & =p_{1,0}+v_{1}t\\ \;\;p_{2,t} & =p_{2,0}+v_{2}t\\ \left|\bar{p_{1,t}p_{2,t}}\right| & =2r \end{array}$$

By solving this system, we obtain the expression shown in Equation (3):(3)$${\left(p_{2,0}+v_{2}t\right)}^{2}-{\left(p_{1,0}+v_{1}t\right)}^{2}=4r^{2}$$
which results in a second-degree equation as a function of t. After being solved, several situations may occur:No real root is obtained. This means that the protected zones of both UAVs do not contact at any time. No conflict situation arises and, therefore, there are not potential collisions.A real root is obtained. This means that the protected zones of both UAVs contact each other without overlapping, which does not generate a conflict situation either.Two real roots, $t_{1}$ and $t_{2}$ are obtained. These values indicate the instants of the beginning and end of the conflict. During this time interval, the protected zones of both UAVs are partially overlapped. If both roots are positive, the conflict is predicted in the future. If only one root is negative, $a_{1}$ and $a_{2}$ are currently in a conflict situation. Finally, if both roots are negative, the conflict was resolved in the past, or it never occurred (the navigation algorithm prevented it).

As an example, Figure 3 shows two UAVs whose trajectories lead to a conflict in the future.

Table 2 presents the implementation of the conflict prediction operation between two UAVs. If the *ConflictPrediction* function is given a velocity, the prediction is performed using this value (lines 04–06). Otherwise, the velocity previously assigned to the UAV will be used (03). We will use this function later to check whether a velocity is suitable for resolving a conflict before assigning it to a UAV. Then, the system of equations is implemented in the variable $t_{c}$ (07–09) and its roots are computed (10). If any real root results (11), $t_{c}$ will be the smallest of them (12). If no real root exists, $t_{c}$ would be assigned to ∞ (14).

To predict whether a particular UAV will be involved in a conflict, we can use the conflict prediction operation just described (see Table 2), using as arguments the UAV under study and the rest of the UAVs present in the airspace. This check can be implemented through a loop, which will be interrupted as soon as the first conflict is detected, to replace the velocity of the UAV analyzed by a new valid velocity. If, on the other hand, this loop concludes without detecting any conflict, the current UAV velocity will be replaced by the direct velocity to the destination. We will discuss all this in more depth in later subsections.

Table 3 implements this behavior using a Boolean function. The main loop (01–08) checks if there is a conflict between each pair of UAVs (05). If so, a *true* value is immediately returned (06). Obviously, we must omit this check for the current UAV (02–04). If, after considering all the UAVs in the airspace, no conflict has been found, a *false* value is provided (09), indicating that there is no conflict with the UAV under study.

### 3.3. Valid Velocity Computation

After predicting a conflict, PCAN replaces the direct velocity of the UAV concerned by a new one that guarantees that no conflicts will occur. Each UAV must have a final velocity before moving during the following $t_{nav}$ time units, which guarantees that it will not collide with any other UAV in the airspace before the next execution of the navigation computation.

A set of candidate velocities “close” to the direct velocity are generated to obtain this final velocity, and these new velocities are checked for conflicts. Suppose any of them manages to resolve all the conflicts between the UAV under study and the rest of the UAVs in the airspace. In that case, this velocity will be a valid velocity, and it will be taken as the final velocity for the UAV. If none of these velocities leads to a conflict-free scenario, the algorithm proceeds to generate a new set of candidate velocity vectors. This process continues until a final velocity is found or a preset limit of iterations is exceeded. If, after exhausting all iterations and discarding all candidate velocity vectors, any valid velocity has been obtained, a decision is made to assign the UAV a zero velocity (assuming it is a rotary-wing UAV). This is done because, in this case, the airspace is very crowded, and no velocity in any direction would allow the UAV to proceed while guaranteeing a conflict-free scenario without increasing the time or distance traveled too much. The UAV should stop moving forward to leave its path clear for other UAVs to move in this situation.

To generate all these candidate velocities, a loop can be used that generates new velocities from the previous ones. In its first iteration, velocities generated are based on the direct one ($v_{d}$). In each iteration, three candidate velocities are generated. One of these velocities will have the same direction as the previous one but a lower modulus. The other two velocities will have a different direction but the same modulus. Through the coef parameter, we can set how much the velocity is reduced when we are varying its modulus. In the same way, through the α parameter (angular displacement), we can control how much the direction will vary when we are obtaining two new velocities with different directions. These two velocities will have the same amount of direction variation, but one will be varied clockwise and the other counterclockwise.

We can see an example in Figure 4. As stated, the iterative process starts with $v_{d}$. The three new candidate velocities generated in iteration i are called $v_{i}^{j}$, where j distinguishes between them. In the figure, we can see that $v_{1}^{1}$ and $v_{1}^{2}$ (orange color) present the same angular displacement with respect to $v_{d}$, but in different directions. On the other hand, $v_{1}^{3}$, which is the modified velocity in modulus, follows the direction of $v_{d}$. If none of these three candidate velocities were valid, we would proceed to generate new ones ($v_{2}^{j}$), following the same methodology, but this time starting from the velocities generated in the previous loop iteration.

Table 4 provides the function that generates valid velocities for a UAV. As described above, three different velocities will be evaluated, which are initialized using the direct velocity (02). The computation is enclosed in a loop (03–17), which iterates until a valid velocity is found or a predetermined maximum number of iterations (m) is reached. At each iteration, three modified velocities are obtained from the previous velocity vector. We use the coef parameter (a value less than 1) to modify the modulus of the velocity vector (04). The change in direction is performed by the *Veer* function (05–06), described below. These three candidate velocities are then checked for conflicts (07, 10, 13). If any of them did not cause any conflict, the corresponding velocity vector is returned (08, 11, 14). If the loop concludes without finding any valid velocity, a null velocity is provided (18).

Table 5 details the behavior of the *Veer* function. The v argument is the velocity vector to vary. The α argument defines the angular displacement between iterations (expressed in radians). In our study, this value has been assigned to the maximum allowed angular displacement divided by the maximum number of iterations (m). The *direction* argument admits the values clockwise and counterclockwise, and represents the direction of the angular displacement (01–03). In (04), a new velocity vector with an angular displacement in a counterclockwise or clockwise direction, respectively, is obtained.

### 3.4. Navigation Computation. The PCAN Algorithm

We have seen the above two mechanisms to predict a conflict between a UAV and the rest of the UAVs in the airspace and provide a new valid velocity for a UAV in that situation. This subsection will describe the general mechanism to compute the final velocity for all the UAVs in the airspace (what we have called navigation computation).

Table 6 details the general behavior of the PCAN algorithm. First, all UAVs are assigned direct velocity to the destination (02–06). Then, for each UAV we check if it produces any conflict with the airspace (08). If there are no conflicts, we assign the direct velocity as the final velocity for the considered UAV (15). However, if a conflict is predicted, a valid velocity is computed for that UAV (09).

If the *ValidVelocity* function returns a null velocity (10), we must recalculate the final velocities for the previously processed UAVs (11). If the velocity is not null, we assign the obtained velocity as the final velocity for the UAV (15).

It is important to clarify that the process must be carried out in sequence, processing one UAV after another until all of them have a final velocity. When a UAV is about to calculate its final velocity, conflict prediction will be made using the final velocities of the UAVs that already have calculated their final velocity and the direct velocities of the UAVs that have not been processed yet. In the case of the first UAV processed, conflict prediction will be made from the direct velocities of the rest of the UAVs, while the last UAV processed will perform the prediction from the final velocities of the rest of the UAVs in the airspace. Since a final velocity for a UAV will only be accepted if it does not lead to a conflict with any other UAV, the order in which the final velocities are obtained will affect the solution for each of the UAVs. The UAV whose final velocity is calculated first will have to fight more conflicts than the last UAV processed, which will not have to fight any conflict, since the previous UAVs treated have already prevented the conflict with it.

For this work, the order in which the UAVs are processed is given by their number within the airspace (ordinal within the set). This implies a priority which, although not chosen, is necessary since there must be an order. However, as future work, we plan to implement a priority system based, for example, on UAV categories (as established by U-space), on the urgency of the service provided, or on any other criteria. In this way, those UAVs with higher priority will minimally modify the optimal trajectory offered by the direct method, while those UAVs with lower priority will have to avoid a greater number of conflicts, obtaining for them a solution farther away from the optimal one.

Finally, note that assigning a null velocity to a UAV implies a higher computational cost because it invalidates the final velocities of UAVs that have been processed before it. By assigning a null velocity, the prediction made by another UAV for which the final velocity was already calculated is invalidated and must be performed again. This process only occurs in really crowded airspaces.

### 3.5. Airspace Bounding

As mentioned before, valid velocities are generated from the airspace state. However, in airspaces with a high density of UAVs, which is expected to happen in UAM scenarios, the computational cost of calculating these valid velocities increases. Moreover, considering UAVs far enough away from a given UAV not to cause a conflict with it in the short term does not make sense and leads to worse solutions. For this reason, an immediate improvement of PCAN would consist of analyzing only the portion of the airspace containing those UAVs that could collide with the UAV understudy before the next execution of the navigation computation.

In short, we define a boundary radius (BR), establishing a circular region around the UAV so that only the UAVs inside that area will be considered for the calculation of the valid velocity. This radius should be greater than the threshold ($BR_{min}$) defined by the worst possible situation, consisting of two UAVs flying in opposite directions, according to:(4)$$BR\geq BR_{min}=2\left(v_{max}t_{nav}+r\right)$$

Figure 5 shows an example. Given the airspace $A=\left\{a_{1}\ldots a_{6}\right\}$, we proceed to calculate a valid velocity for $a_{1}$. Consequently, only the UAVs within the circular region defined by BR, plotted in green, are considered for the valid velocity calculation. Thus, $a_{5}$ and $a_{6}$ do not influence the final velocity of $a_{1}$.

## 4. Performance Evaluation

In this section, we present the performance evaluation results of the proposed algorithm (PCAN), with the airspace bounding improvement described in Section 3.5. This evaluation has been carried out using simulation techniques.

The simulation tool employed has been developed in Matlab R2020b [28] by making use of the object-oriented programming language offered by this platform. The simulator allows the configuration of the parameters of the proposed algorithm and the generation of random scenarios of any size. It also provides result reports, generating plots such as the ones shown below.

In total, more than 150,000 scenarios have been simulated, with different configurations. As we will see next, the PCAN algorithm has prevented the occurrence of conflicts in all cases.

### 4.1. Two-UAV Study

First, we will analyze the behavior of PCAN when $A=\left\{a_{1},a_{2}\right\}$. Figure 6 shows the different scenarios evaluated. Given a circumference of 4 km in diameter, in all the scenarios $a_{1}$ starts its flight from the west (point p), crosses the circumference passing through its center, and ends at the opposite point of the circumference (w). On the other hand, $a_{2}$ starts its flight from a different position in each scenario. In particular, we have considered 18 different initial positions for $a_{2}$ ($p_{1}\ldots p_{18}$) resulting in 18 relative angles between both UAVs (between 0° and 170°, in 10° intervals). Each initial position ($p_{1}\ldots p_{18}$) has an associated destination ($w_{1}\ldots w_{18}$), which is also reached by passing through the center of the circumference. In each configuration, $a_{1}$ and $a_{2}$ start the flight simultaneously, with $v_{max}=13.9\;m/s$ (50 km/h), finally colliding at the center of the circumference if the navigation algorithm does not prevent it. We have set r=45 m, $t_{nav}=3\;s$ and τ=0.3 s, and PCAN parameters as BR=∞, m = 50, coef=0.95, and $\alpha =\frac{\pi }{m}\;\mathrm{rad}$. Moreover, the safety radius r has been increased 8.34 m, as Figure 1 shown explains.

Figure 7 shows a scenario number corresponding to each of the 18 configurations described above on its abscissa axis. The ordinate axis represents the number of conflicts produced in each scenario. The two series represent the results obtained by the direct and PCAN algorithms. As expected, the direct algorithm produces conflicts (leading to collisions) in all scenarios. In contrast, the PCAN algorithm manages to avoid them in all the scenarios analyzed.

As detailed, PCAN avoids conflicts by causing UAVs to deviate from the optimal trajectory to the destination or by reducing their velocity (but maintaining the optimal trajectory). This can lead to increases in the distance traveled by the UAVs and the time required to reach their destinations. Figure 8a shows the average distance traveled by the UAVs, expressed in meters. The direct algorithm indicates the minimum distance between the initial and the final position. The increase due to the deviations made by PCAN to avoid conflict is negligible in this plot. In the case of the BBCA algorithm, an increase in the distance traveled is clearly visible.

To better analyze these results, Figure 8b shows the increase in the distance traveled by the UAVs when using PCAN and BBCA relative to the minimum distance provided by the direct algorithm. As we can see, the penalty in the distance for BBCA is appreciable in some scenarios. In PCAN, this increase is negligible again (about 0.11% in the worst case, but about 0.06% in most of the scenarios).

We have also analyzed the impact on the flight time due to the avoidance maneuvers performed by the UAVs. Figure 9a shows the average flight time of UAVs, expressed in seconds. In the case of PCAN, it can be observed that the increase in time is greater than in the distance. This is because PCAN tends to modify the velocity modulus (instead of the UAV trajectory) to avoid the conflict, thereby slightly increasing the time the UAV remains in flight.

Analogous to Figure 8b, Figure 9b shows the relative increase (with respect to the direct algorithm) in the UAV flight time when using PCAN and BBCA. For PCAN, we can observe an increase of less than 2% in all the scenarios. One more time, the penalty introduced by BBCA in some scenarios is notable.

### 4.2. Multi-UAV Study

Next, we study the behavior of PCAN when it is used in airspaces with multiple UAVs. We have considered a 5×5 km flight region and with different UAV densities. In particular, |A|={10, 20, 30, 40, 50, 60, 70, 80, 90, 100}. Figure 10 shows an example. For each density, 24 random route configurations have been generated for the UAVs, considering that the initial and the final position cannot be less than 100 m from the edges of the region and less than 1 km away. Each route configuration has been simulated with the direct, BBCA, APF, and PCAN algorithms, resulting in 960 runs. In all cases, we have set $v_{max}=50\;\mathrm{km}/h$, r=40 m, $t_{nav}=3\;s$ and τ=0.3 s, and PCAN parameters as $BR=3v_{max}t_{nav}+2r$, m=50, coef=0.95, and $\alpha =\frac{\pi }{m}\;\mathrm{rad}$. Moreover, the safety radius r has been increased 8.34 m, as Figure 1 explains.

The abscissa axis in the plot of Figure 11a indicates the number of UAVs in the flight region. In contrast, the ordinate axis shows the mean number of conflicts produced and the corresponding standard deviation. For the direct method, conflicts increase exponentially with the number of UAVs in the region. The APF algorithm manages to resolve isolated conflicts, but its performance decreases as the UAV density increases. BBCA significantly reduces the number of conflicts but does not eliminate them. Finally, as expected, PCAN successfully resolves all the conflicts.

Figure 11b shows the same results, but now from the point of view of the percentage of solved conflicts. We can clearly see that PCAN and BBCA outperform APF. Obviously, the direct method is not plotted since it does not avoid any conflict.

Figure 12a shows the distance traveled by the UAVs, in absolute terms, as a function of the UAV density. As expected, when applying the direct method, the distance traveled by each UAV does not vary with its number, while the standard deviation decreases. On the contrary, the detours introduced by BBCA increase the distance linearly. Finally, PCAN also increases the distance traveled by the UAVs, but to a lesser extent since it resolves the conflicts more efficiently.

In Figure 12b, we can see the relative increase in the distance traveled by the UAVs regarding the direct algorithm. We can observe the linearity in this increase, which in the case of PCAN progresses from 0% to about 4% for the 100-UAV configuration. If we compare PCAN to BBCA, we can see that the increase in distance is marginal, even avoiding all conflicts. APF performs slightly better than PCAN, but at the cost of not being conflict-free.

Figure 13a shows the flight time of UAVs, in absolute terms, as a function of the UAV density. For the direct method, flight time does depend on the number of UAVs. As shown, PCAN offers better results than the rest of conflict management algorithms.

Finally, Figure 13b compares the flight time for BBCA, APF, and PCAN with respect to the direct algorithm. In the case of PCAN, in the highest density scenarios, the flight time is increased by about 6%, outperforming, in any case, the other two techniques.

### 4.3. Computation Time

To conclude the analysis, we studied the time used by the direct, PCAN, BBCA, and APF algorithms to offer their solutions. Figure 14 shows the time consumed to produce an output for every UAV each time the navigation computation is executed, expressed in milliseconds. Results were generated with an Intel i9-10900KF@3.7GHz processor, using one core.

As shown, PCAN works very well in low-density scenarios exhibiting better behavior than BBCA from 10 to 50 UAVs. As the density of UAVs increases, PCAN requires more time to solve all the potential conflicts produced, which were shown in Figure 11a, series *direct*. In any case, the computation time does not represent a dramatic bottleneck. For the 100-UAV configuration, the time employed is less than 0.9 ms. Note that these are extremely dense scenarios in which avoiding all conflict is complex.

## 5. Conclusions and Future Works

This paper proposes the PCAN algorithm for conflict-free navigation among a set of UAVs flying over urban airspace according to a set of predetermined flight plans. In each run of the algorithm, assuming that all UAVs are heading in a straight line and at maximum velocity towards their destinations, the algorithm predicts future conflicts between them and proposes modifications in their velocities to prevent their occurrence. The analysis carried out shows that PCAN results, in the worst case, in an increase in the distance traveled by the UAVs of about 4% and an increase in flight time of approximately 6%. This makes our conflict avoidance proposal suitable for the deployment of tactical conflict resolution services in the framework of the future U-Space UAM space.

As future work, we plan to revise the decision-making of PCAN to reduce the overhead involved in conflict avoidance. Instead of performing an intensive search of modified velocities, the algorithm can start by exploring the most promising velocities according to the airspace state. Moreover, as discussed in Section 3.4, we can implement a priority system based on categories. In this way, those UAVs with higher priority will see their trajectories modified to a lesser extent. Finally, we must consider the existence of geofences in the airspace that define no-fly zones.

## Figures and Tables

**Figure 1 sensors-21-05649-f001:**
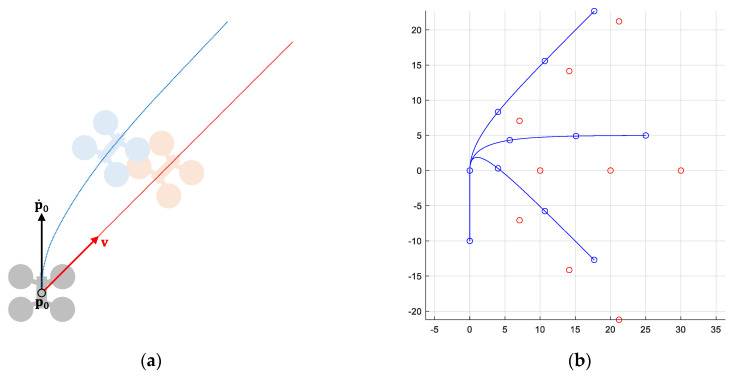
UAV dynamic model. (**a**) Example of a UAV turning 45° clockwise: the red line shows the ideal trajectory followed if the commanded velocity is applied instantaneously ($\left|{\dot{p}}_{0}\right|=\left|v\right|=10\;m/s$); the blue line shows its dynamic trajectory assuming τ=0.5 s. (**b**) Ideal and real position (every second) of the UAV when turning 45°, 90°, and 135°, respectively.

**Figure 2 sensors-21-05649-f002:**
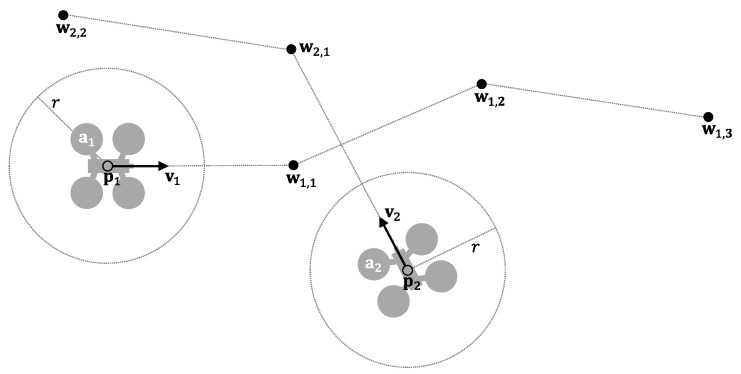
Two UAVs in the airspace.

**Figure 3 sensors-21-05649-f003:**
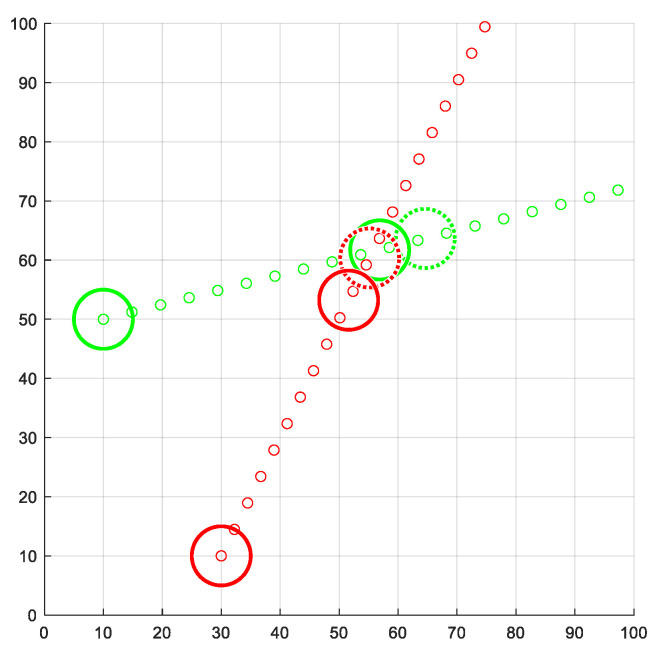
Example of conflict prediction (r=5 m; v=10 m/s). The two UAVs maintain a conflict from t=4.83 s (solid lines) to t=5.63 s (dotted lines).

**Figure 4 sensors-21-05649-f004:**
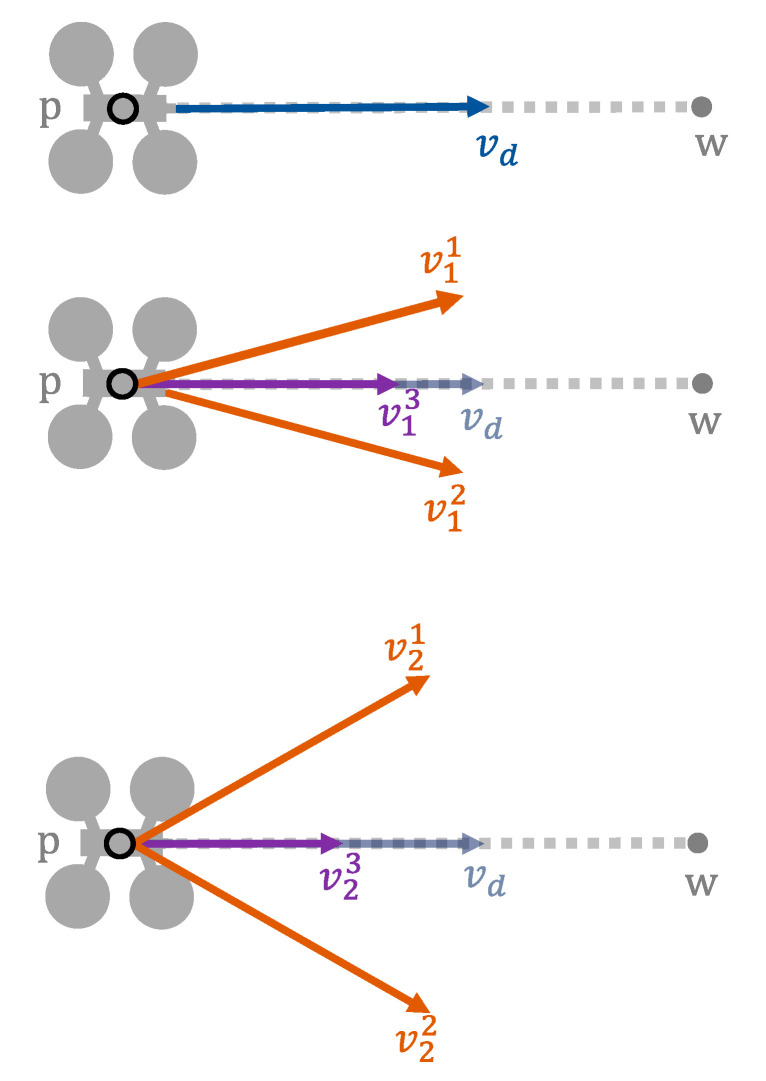
Searching for a valid velocity from the direct one. Above, a UAV with direct velocity ($v_{d}$) flying from the initial position (p) to the destination (w). In the center, three candidate velocity vectors have been generated in the first iteration. Below, new candidate velocity vectors have been generated, based on those of the previous iteration (if they did not lead to a conflict-free scenario).

**Figure 5 sensors-21-05649-f005:**
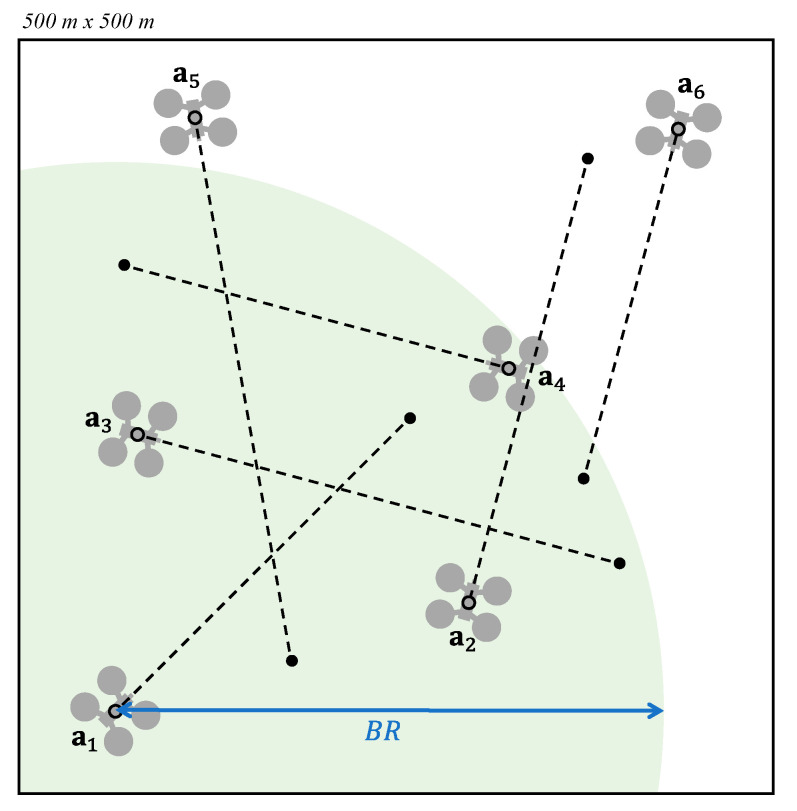
Airspace bounding for $a_{1}$.

**Figure 6 sensors-21-05649-f006:**
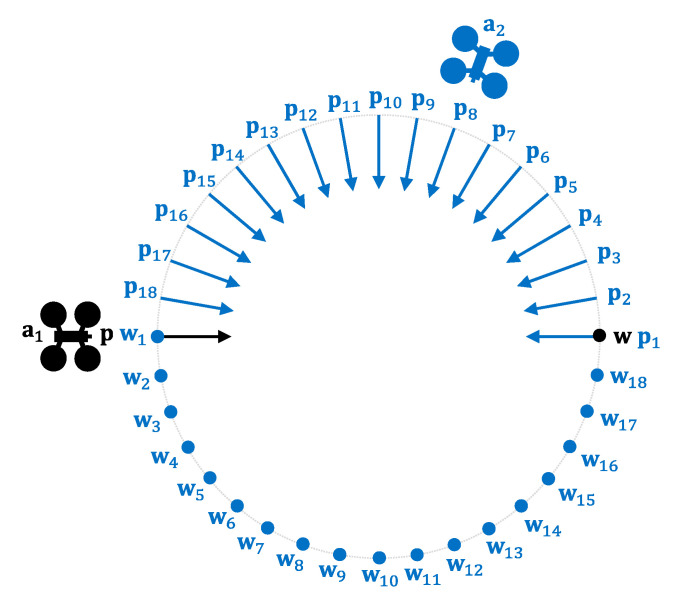
Two-UAV scenarios: relative angles between trajectories.

**Figure 7 sensors-21-05649-f007:**
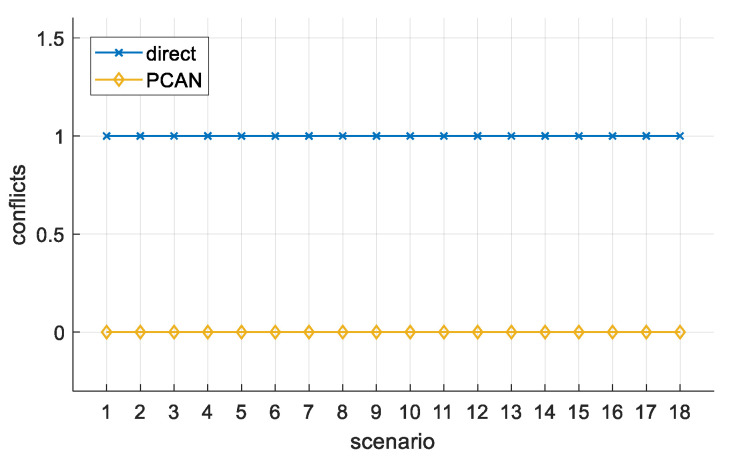
Two-UAV study. The number of conflicts.

**Figure 8 sensors-21-05649-f008:**
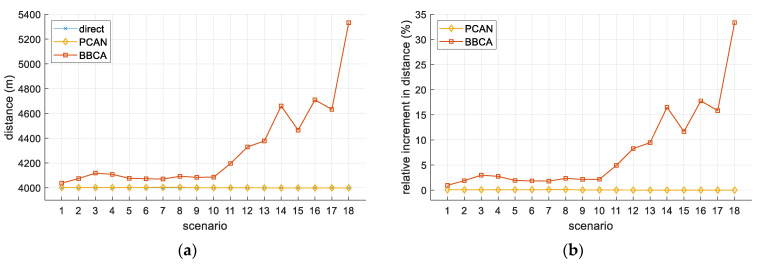
Two-UAV study. Distance traveled: (**a**) absolute values; (**b**) relative increase with respect to the direct algorithm.

**Figure 9 sensors-21-05649-f009:**
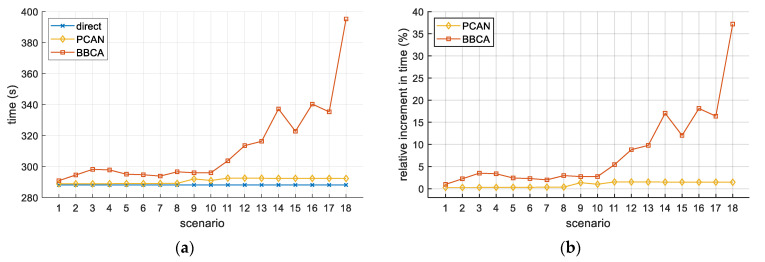
Two-UAV study. Flight time: (**a**) absolute values; (**b**) relative increase with respect to the direct algorithm.

**Figure 10 sensors-21-05649-f010:**
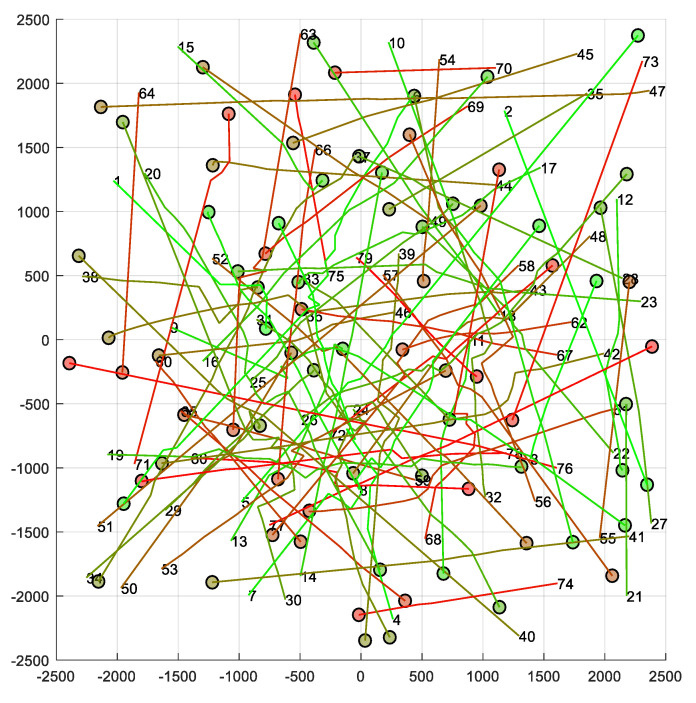
Random scenario with 80 UAVs and PCAN. Each number represents the initial position of a UAV; each circle represents the current position of the UAV and its safety radius; each line represents the trajectory followed by the UAV. Units on the X and Y axes are meters.

**Figure 11 sensors-21-05649-f011:**
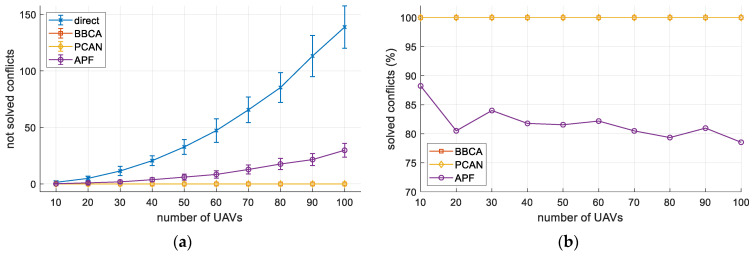
Multi-UAV study. The number of conflicts: (**a**) not solved conflicts (mean and standard deviation); (**b**) solved conflicts (%).

**Figure 12 sensors-21-05649-f012:**
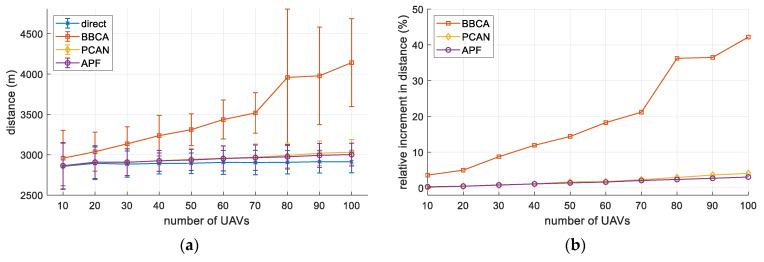
Multi-UAV study. Distance traveled: (**a**) absolute values (mean and standard deviation); (**b**) relative increase with respect to the direct algorithm.

**Figure 13 sensors-21-05649-f013:**
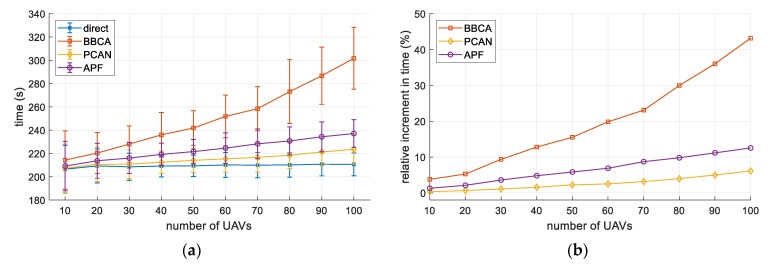
Multi-UAV study. Flight time: (**a**) absolute values (mean and standard deviation); (**b**) relative increase with respect to the direct algorithm.

**Figure 14 sensors-21-05649-f014:**
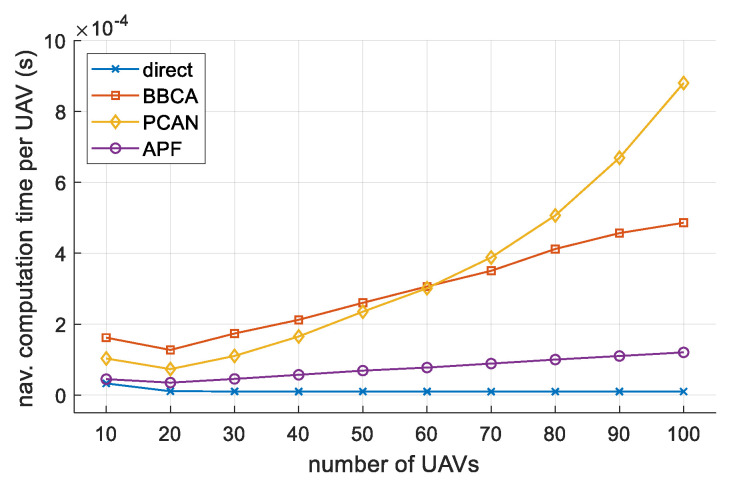
Time (per UAV) consumed by the navigation algorithm.

**Table 1 sensors-21-05649-t001:** Direct navigation to the destination.

	v=DirectNav(a)
01	[p v {w, R}]=a
02	if $\left|\bar{pw}\right|<v_{\max}t_{\mathrm{nav}}\;\&\;R\neq \emptyset$
03	a=[p v R]
04	{w, ~}=R
05	end if
06	$v=\frac{\bar{pw}}{\left|\bar{pw}\right|}v_{\max}$

**Table 2 sensors-21-05649-t002:** Conflict prediction between two UAVs.

	$t_{c}=ConflictPrediction\left(v,\;a_{1},a_{2}\right)$
01	assume r
02	$\left[p_{1}v_{1}\sim \right]=a_{1}$
03	$\left[p_{2}v_{2}\sim \right]=a_{2}$
04	if **v** ~= null
05	$v_{2}=v$
06	end if
07	$t_{2}={\left(v_{1}-v_{2}\right)}^{2}$
08	$t_{1}=2\left(p_{1}-p_{2}\right)\left(v_{1}-v_{2}\right)$
09	$t_{0}={\left(p_{1}-p_{2}\right)}^{2}-4r^{2}$
10	$t_{c}=\mathrm{roots}\left(\left[t_{2}t_{1}t_{0}\right]\right)$
11	if $\mathrm{isreal}\left(t_{c}\right)\;\&\;t_{c}>0$
12	$t_{c}=\min\left(t_{c}\right)$
13	else
14	$t_{c}=\infty$
15	end if

**Table 3 sensors-21-05649-t003:** Conflict prediction between a UAV and the rest.

	$boolean=\mathrm{AirspaceConflicts}\left(v,\;a_{1},A\right)$
01	for all $a_{2}\;\in \;A$ do
02	if $a_{1}=a_{2}$
03	continue
04	end if
05	if $\mathrm{ConflictPrediction}\left(v,\;a_{1},a_{2}\right)=\infty$
06	return *true*
07	end if
08	end do
09	return *false*

**Table 4 sensors-21-05649-t004:** Computation of a valid velocity for a UAV.

	$v=\mathrm{ValidVelocity}\left(a_{1},A\;\right)$
01	assume m, coef, α
02	$v_{1}=v_{2}=v_{3}=\mathrm{DirectNav}\left(a_{1}\right)$
03	while *m* > 0
04	$v_{1}=v_{1}*coef$
05	$v_{2}=\mathrm{Veer}\left(v_{2},\alpha ,counterclockwise\right)$
06	$v_{3}=\mathrm{Veer}\left(v_{3},\alpha ,clockwise\right)$
07	if $\mathrm{not}\;\mathrm{AirspaceConflicts}\left(v_{1},a_{1},A\right)$
08	return $v=v_{1}$
09	end if
10	if $\mathrm{not}\;\mathrm{AirspaceConflicts}\left(v_{2},a_{1},A\right)$
11	return $v=v_{2}$
12	end if
13	if $\mathrm{not}\;\mathrm{AirspaceConflicts}\left(v_{3},a_{1},A\right)$
14	return $v=v_{3}$
15	end if
16	*m* = *m* − 1
17	end while
18	return **v** = **0**

**Table 5 sensors-21-05649-t005:** Velocity vector direction change.

	$v^{\prime }=\mathrm{Veer}\left(v,\;\alpha ,\;\mathrm{direction}\right)$
01	if direction=clockwise
02	α=−α
03	end if
04	$v^{\prime }=v\times \left[\begin{array}{cc} \cos\left(\alpha \right) & \mathrm{sen}\left(\alpha \right)\\ -\mathrm{sen}\left(\alpha \right) & \cos\left(\alpha \right) \end{array}\right]$

**Table 6 sensors-21-05649-t006:** The PCAN algorithm.

	A=PCAN(A)
01	assume r
02	for all a∈A do
03	[p ~ R]=a
04	v=DirectNav(a)
05	a=[p v R]
06	end do
07	for all $a_{i}\in A$ do
08	if $\mathrm{AirspaceConflicts}\left(v,a_{i},\;A\right)$
09	$v=\mathrm{ValidVelocity}\left(a_{i},A\right)$
10	if v=0
11	$A=\mathrm{PCAN}\left(\left\{a_{1}\ldots a_{i-1}\right\}\right)\cup^{\;}\left\{a_{i}\ldots a_{n}\right\}$
12	end if
13	end if
14	$\left[p\;\sim \;R\right]=a_{i}\;$
15	$a_{i}=\left[p\;v\;R\right]$
16	end do

## Data Availability

The data presented in this study are openly available in FigShare at http://doi.org/10.6084/m9.figshare.15830565.

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
