# Peer review of "A Tactical Conflict Resolution Proposal for U-Space Zu Airspace Volumes"

_sensors, 2021, doi:10.3390/s21165649_

Round 1

Reviewer 1 Report

This paper proposes a prediction-based conflict-free adaptive navigation in UAM scenarios. The topic is attractive from a practical point of view for the future air traffic management system. However, the assumption of the UAVs’ dynamics is unrealistic to be verified with an actual application. The detailed review comments are as follows:

  1. In section 3, the authors assume that UAVs are flying to the waypoint directly, but when UAVs are reached the specific range of their waypoint, the targeted waypoint is switched to the next waypoint. It may cause a curved flight path, and the proposed conflict prediction mechanism is not applicable.

  1. The proposed conflict prediction is required to keep the constant speed and heading angle, but it is not possible due to the external disturbance. Also, the speed can be a control parameter which is frequently varying in the time domain. Therefore, the assumption about the constant speed can be a considerable drawback of the proposed method since it can only be applied for constant or slowly moving vehicles.

  1. The proposed method considers a 2D plane, but the altitude separation can be an optimal solution to prevent a collision for vertical takeoff and landing UAVs.

  1. The UAVs dynamics are too simple to be applicable for a real problem. A higher fidelity of the dynamics is required for applying to practical cases.

Author Response

1. In section 3, the authors assume that UAVs are flying to the waypoint directly, but when UAVs are reached the specific range of their waypoint, the targeted waypoint is switched to the next waypoint. It may cause a curved flight path, and the proposed conflict prediction mechanism is not applicable.

As we state in the manuscript, in this work we assume that UAVs move in an airspace without restrictions (such as buildings, geofences, etc.). Therefore, in the absence of conflicts, they could fly in a straight line. In other words, the path to be followed only includes the destination waypoint. U-space contemplates geofences as a means to define restricted areas in the airspace. One way to implement a navigation route between origin and destination avoiding such areas is by means of a sequence of waypoints. However, this solution limits the possibilities for conflict management. Our algorithm proposes that the UAV moves away from the established route in order to avoid conflicts with other UAVs (while maintaining a flexible behavior in this sense). Transit between waypoints under these conditions raises many questions, as the reviewer rightly mentions in his comment. Consequently, as future work, we plan to incorporate geofences into the navigation system, instead of sequences of waypoints. These geofences will prevent those speeds proposed by the navigation system that results in an invasion of restricted zones being considered.

2. The proposed conflict prediction is required to keep the constant speed and heading angle, but it is not possible due to the external disturbance. Also, the speed can be a control parameter which is frequently varying in the time domain. Therefore, the assumption about the constant speed can be a considerable drawback of the proposed method since it can only be applied for constant or slowly moving vehicles.

The algorithm assumes certain behavior for UAVs during a time interval. As the reviewer states, such expected behavior may not be met due to the aircraft dynamics or external disturbances. To support the first factor, a safety margin is applied to the proposed conflict radii and velocity vectors. In this work, the presence of external factors (such as, for example, positioning instrument reading errors or the presence of gusty winds) is not considered. In any case, we have increased the frequency of execution of the navigation algorithm to detect and correct such situations as soon as possible.

3. The proposed method considers a 2D plane, but the altitude separation can be an optimal solution to prevent a collision for vertical takeoff and landing UAVs.

We agree with the reviewer that, in theory, using height would greatly simplify conflict management. However, we have postponed this aspect for future work. The main reason is that, unlike ordinary airspace, Zu airspace volumes defined in U-Space are very limited in height (120 meters above ground level). A straightforward conversion of the current safety circles into spheres leaves no room for multiple levels unless we reduce safety margins to a few meters. We consider it essential that the relevant regulations define limits that allow 3D conflict management to be implemented.

4. The UAVs dynamics are too simple to be applicable for a real problem. A higher fidelity of the dynamics is required for applying to practical cases.

As the reviewer states, it is not at all realistic for the UAV to instantaneously implement a command coming from the navigation system. According to this comment, we have incorporated into our model the dynamic behavior of the UAVs, described in section 3.1. Now, course changes are executed gradually. In order to assume the errors due to such dynamic behavior, we have incorporated to the algorithm certain safety margins applied to the radius of the protected zones and the proposed velocity vectors, together with a reduction of the navigation time (tnav). We have repeated all the simulations with the new parameters, and have updated the plots in the manuscript.

Reviewer 2 Report

General comments
==

Authors aim to solve a very important problem of conflict resolution in future UAV/UAM environment. The proposed algorithm is relatively simple. Number of assumptions made by the authors make it difficult for real implementation (2d space, centralized control, relatively large reaction time tnav, etc.). However authors may improve it by introduction of UAV dynamics into calculation of some important parameters. 

Specific comments
==

For the introduction section, please add UAM / UAV concept of operations proposed by other countries as well. For example, Korean ConOps http://kada.konkuk.ac.kr/2021/06/09/urban-air-mobility-concepts-of-operations/

It is better to combine the section 1 and 2. Introduction typically explains the background of the problem and existing solutions. Conclusion of the introduction section should be the disadvantages of existing methods and need for a new approach that authors propose in the paper.

Section 3.3 authors mention generation of a new set of candidate velocities. Details on calculation of these velocities should be provided. How is the alpha calculated, how is the speed reduction calculated? These parameters are closely dependent on t_nav and UAV's inertial properties.

Section 4.1. tnav=10s are too large numbers. This may produce a large uncertainty in velocity and position estimation. Authors should provide additional estimation for relationship between tnav, r, and vmax. 

In the introduction section, authors mention about computational efficiency of the algorithm. However no comparison with other algorithms is provided in terms of computational time.

Authors also mention that algorithm works for both, centralized control (command center controls all the UAV) and individual control (only single UAV is controllable). Case study for individual control scenario is missing.

Author Response

For the introduction section, please add UAM / UAV concept of operations proposed by other countries as well. For example, Korean ConOps http://kada.konkuk.ac.kr/2021/06/09/urban-air-mobility-concepts-of-operations/

Thanks for the suggestion. To give a broader view we have included the Korean and Australian UAM ConOps in the new version of the manuscript.

It is better to combine the section 1 and 2. Introduction typically explains the background of the problem and existing solutions. Conclusion of the introduction section should be the disadvantages of existing methods and need for a new approach that authors propose in the paper.

We fully agree with the reviewer that the introduction section should clearly state the issues of the previous methods and the advantages of the new one. As suggested, we have revised the introduction section, in order to highlight the contribution to the state-of-the-art of the new technique. In our modest opinion, section 2 contains too many details on previous techniques, which worsen the overall view that the introductory section should offer, justifying its existence.

Section 3.3 authors mention generation of a new set of candidate velocities. Details on calculation of these velocities should be provided. How is the alpha calculated, how is the speed reduction calculated? These parameters are closely dependent on t_nav and UAV's inertial properties.

According to this comment, we have fully rewritten the entire paragraph to clarify these parameters and their meaning:

"Table 5 details the behavior of the Veer function. The v argument is the velocity vector to vary. The α argument defines the angular displacement between iterations (expressed in radians). In our study, this value has been assigned to the maximum allowed angular displacement divided by the maximum number of iterations (m). The direction argument admits the values clockwise and counterclockwise, and represents the direction of the angular displacement (01-03). In (04), a new velocity vector with an angular displacement in counterclockwise or clockwise direction, respectively, is obtained."

Section 4.1. tnav=10s are too large numbers. This may produce a large uncertainty in velocity and position estimation. Authors should provide additional estimation for relationship between tnav, r, and vmax.

The reviewer is right; particularly, in the revised version of the manuscript, which includes the dynamic behavior of the UAVs. For this reason, tnav has been reduced to 3s. Additionally, we have modified sections 3.5, 4.1, and 4.2 to clarify the relationship between these parameters.

In the introduction section, authors mention about computational efficiency of the algorithm. However no comparison with other algorithms is provided in terms of computational time.

According to the reviewer comment, a new section has been added explaining and comparing the computational requirements of each algorithm.

Authors also mention that algorithm works for both, centralized control (command center controls all the UAV) and individual control (only single UAV is controllable). Case study for individual control scenario is missing.

Since in the current work we have focused on evaluating the algorithm in a centralized scenario (assuming that this is a conflict management service offered by a U-Space service provider), and to avoid misunderstanding, we have removed from the Introduction section the mention of the possibility of executing it in a distributed way, which we leave as future work. As we stated, this alternative requires at all times a global knowledge of the situation on the part of each UAV.

Round 2

Reviewer 1 Report

  1. The assumption that only one waypoint in a path is a critical drawback of the proposed method, and there are many works of literature dealing with a complex flight path. In this point of view, the novelty of the proposed method is questionable to be published as a journal paper. A logical background of the one waypoint assumption in the practical point of view is necessary for showing the worth of the proposed method.
  2. The safety margin is not a good solution to cover the low fidelity dynamic model since the safety margin is defined by a heuristic method that doesn’t have any physical meaning. An analysis, which depicts the safety margin can replace the role of the detailed dynamics model, is necessary.
  1. According to the small UAS regulation, which is published by FAA, the maximum allowable altitude is 400 feet (121 m) above the ground for the small UAVs. It means that 120 meters are enough to operate small UAVs. Still, collision avoidance without the altitude separation in a zone which is having 120 altitudes is not a fancy solution.
  1. The dynamic behaviour of the UAVs, which is depicted in section 3.1, looks like a typical actuating system with the first-order time lag, and it is unusual to express the dynamics of the UAVs. Some literature survey that depicts the UAVs’ dynamics is required to understand the UAVs’ dynamics, and a reasonable dynamics model is still required to upgrade the quality of the paper.

Author Response

The assumption that only one waypoint in a path is a critical drawback of the proposed method, and there are many works of literature dealing with a complex flight path. In this point of view, the novelty of the proposed method is questionable to be published as a journal paper. A logical background of the one waypoint assumption in the practical point of view is necessary for showing the worth of the proposed method.

As suggested by the reviewer, we have added to the navigation mechanism the tracking of paths composed of multiple waypoints. It has required the modification of Figure 1 (now Figure 2) and tables 1 and 6 (tables 2 and 4 have also been modified to enrich matrix notation). We have repeated the simulations and updated accordingly all the results shown in the evaluation section of the manuscript.

In the first comment the reviewer did, he/she asked about conflict management during the transit between consecutive waypoints. In this regard, note that the navigation mechanism is executed periodically, and we have implemented that the UAV switches to the next waypoint when it would reach the current one before the next execution. As conflict management is a memoryless process transparent to course changes (in other words, it adapts to such changes regardless of whether they are necessary to avoid a collision or to perform a transition between waypoints), the incorporation of multiple-waypoint paths does not affect it.

The safety margin is not a good solution to cover the low fidelity dynamic model since the safety margin is defined by a heuristic method that doesn’t have any physical meaning. An analysis, which depicts the safety margin can replace the role of the detailed dynamics model, is necessary.

Indeed. As the reviewer states, the safety margin initially applied had not have a physical meaning. In the new version of the manuscript, we have analyzed the system dynamics and bounded the maximum error produced in the tracking of the ideal reference. We have incorporated this knowledge into the conflict management process to prevent conflicts from occurring due to this cause. The results obtained in the simulations confirm this hypothesis.

Moreover, this has allowed us to detect and correct an error that we made in the previous revision, external to the algorithm, which increased the calculations performed by the simulator above those necessary, as can be seen in the current Figure 14.

According to the small UAS regulation, which is published by FAA, the maximum allowable altitude is 400 feet (121 m) above the ground for the small UAVs. It means that 120 meters are enough to operate small UAVs. Still, collision avoidance without the altitude separation in a zone which is having 120 altitudes is not a fancy solution.

According to the reviewer’s comment, we have mentioned in Section 3 that “it is possible to run the algorithm in parallel, by using multiple layers at different heights. The criteria for the choice of layer are left to the U-Space service provider, and can be very varied (UAV heading, priority, maneuverability, airspace congestion...). The only requirement is that, during vertical takeoff and landing, an UAV must participate in the conflict management of the layers below the deployment one.”

The dynamic behaviour of the UAVs, which is depicted in section 3.1, looks like a typical actuating system with the first-order time lag, and it is unusual to express the dynamics of the UAVs. Some literature survey that depicts the UAVs’ dynamics is required to understand the UAVs’ dynamics, and a reasonable dynamics model is still required to upgrade the quality of the paper.

As the reviewer suggests, we have cited a previous work that describes in detail the dynamics of a quadcopter-type UAV (including 3D translation and rotation motions), as well as the low-level control system that controls the UAV motors and allows the tracking of a given reference velocity vector. In this way, the reader can fully understand the dynamics of the UAV. With that starting point, the conflict management algorithm uses a simplified dynamics (2D translation). As mentioned above, the explanation of such dynamics has been greatly improved, including behavioral examples and error bounding (see Figure 1 in the new version of the manuscript).

We want to thank again to the reviewer for his/her comments, which have undoubtedly contributed to a substantial improvement of our proposal.

Reviewer 2 Report

The authors have fully answered to all the comments I have made previously. I think the paper can be considered for publication.

Authors should carefully redraw figures 6, 7, 8, 10, 11

  • axis title must include the description or variable name in mathematical notation. Not variable name as it appears in a source code. Example numUAVs -> number of UAVs; rel inc dist -> ...
  • axes should be adjusted to fill the whole figure. For example in Figure 8a, there is no need for y-axis range to start from 0. Set the range from 250 to 350 to fill the most of the space.

Author Response

Authors should carefully redraw figures 6, 7, 8, 10, 11

axis title must include the description or variable name in mathematical notation. Not variable name as it appears in a source code. Example numUAVs -> number of UAVs; rel inc dist -> ...

axes should be adjusted to fill the whole figure. For example in Figure 8a, there is no need for y-axis range to start from 0. Set the range from 250 to 350 to fill the most of the space.

We have revised the axis titles and ranges in from figure 6 to figure 14. We sincerely thank the reviewer for his/her comments, which have undoubtedly contributed to improve the quality of the manuscript.